# The Endoplasmic Reticulum: A Central Hub for Dermal Collagen Quality Control and Its Profound Implications for Skin Health and Disease

## Abstract

The integrity and functional resilience of the skin are critically dependent on the extracellular matrix (ECM), particularly its abundant collagen network. Dermal collagen undergoes intricate biosynthesis within dermal fibroblasts, with the Endoplasmic Reticulum (ER) serving as the primary site for folding, assembly, and post-translational modification. The ER houses a sophisticated quality control system that monitors the quantity and precise quality of procollagen molecules, involving molecular chaperones, folding enzymes, and post-translational modification machinery such as Hsp47, protein disulfide isomerase (PDI), prolyl hydroxylases, and lysyl hydroxylases. This review delineates the critical ER-centric mechanisms governing dermal procollagen biogenesis and quality surveillance, highlighting how disruptions can lead to ER stress, activate the Unfolded Protein Response (UPR), and contribute to skin conditions like aging, scarring, and fibrotic disorders. By consolidating current knowledge on ER's role in dermal collagen homeostasis, we aim to underscore its significance as a therapeutic target for maintaining skin health and ameliorating collagen-related dermatological pathologies.

Keywords: Dermal collagen, Endoplasmic Reticulum, Quality Control, Hsp47, Prolyl Hydroxylation, Lysyl Hydroxylation, ER Stress, Unfolded Protein Response, Skin Aging, Fibrosis.

## 1 Introduction

The skin's robust mechanical properties, including strength, elasticity, and resilience, are largely attributed to the dermal extracellular matrix (ECM), predominantly composed of collagen. Type I collagen constitutes 80-90% of dermal collagen, providing tensile strength, while Type III collagen (10-15%) contributes elasticity. Minor collagens, such as Type V, further refine fibril organization. This complex network is essential for skin integrity, wound healing, and resisting mechanical stress.

Collagen biosynthesis is a complex and highly regulated process initiated within dermal fibroblasts. The journey begins with the translation of pro-alpha collagen chains and their translocation into the Endoplasmic Reticulum (ER), the primary organelle for its folding, assembly, and extensive post-translational modifications (PTMs). The ER acts as a sophisticated, multi-layered quality control system, actively determining the quantity and precise quality of each procollagen molecule. This ER-centric surveillance is paramount, as the secretion of misfolded or improperly modified procollagen compromises dermal matrix integrity, leading to weakened, unstable, or non-functional collagen fibrils. Such deficiencies are implicated in various skin conditions, from aging and reduced elasticity to impaired wound healing, keloid formation, and fibrotic disorders. This review examines the critical ER-based mechanisms governing dermal procollagen biosynthesis and quality control, exploring the

Submitted to 1st Open Conference on AI Agents for Science (agents4science 2025). Do not distribute.

consequences of their failure, and highlighting their implications for skin health and dermatological pathologies.

## 2   Intracellular Quality Control of Procollagen in Dermal Fibroblasts

The Endoplasmic Reticulum (ER) serves as the primary and indispensable checkpoint for the folding and quality control of procollagen. This rigorous system ensures that only correctly folded and assembled procollagen trimers exit for further processing and deposition into the extracellular matrix. Misfolded or aberrant molecules are retained, subjected to refolding cycles, and eventually degraded if defects persist. This active surveillance within the ER directly determines the structural integrity and biological efficacy of dermal collagen, profoundly influencing skin's biomechanical properties and susceptibility to pathology. Maintaining ER quality control efficiency is paramount for dermal proteostasis and skin health

### 2.1   ER as the Primary Site for Procollagen Folding and Maturation

Procollagen biosynthesis begins with mRNA translation on ER-bound ribosomes and co-translational translocation of nascent polypeptide chains into the ER lumen. The ER's oxidizing environment facilitates disulfide bond formation, particularly in the C-terminal propeptide domains, which guide the correct association of three pro-alpha chains into a procollagen trimer. Simultaneously, the ER provides a specialized milieu with enzymes and chaperones that catalyze and monitor extensive post-translational modifications (PTMs), crucial for procollagen maturation. Successful and timely execution of these events is essential; deviations lead to misfolded procollagen retention, ER stress, or secretion of functionally compromised molecules, ultimately impairing dermal ECM structure and function.

### 2.2   Key Molecular Chaperones and Folding Enzymes in Dermal Fibroblasts

The precise folding and assembly of dermal procollagen are meticulously orchestrated by a specialized network of molecular chaperones and enzymes within the ER lumen of dermal fibroblasts. These components prevent aggregation, ensure correct triple helix association, and facilitate critical post-translational modifications. This system actively surveils procollagen quality, ensuring only functional collagen integrates into the dermal matrix.

#### 2.2.1   Hsp47: Specific Chaperone for Dermal Procollagen Triple Helix Formation

Heat Shock Protein 47 (Hsp47), or serpin H1, is a collagen-specific ER-localized molecular chaperone. Its consistent co-expression with collagen highlights its role in collagen proteostasis. Hsp47 specifically binds to nascent triple-helical domains, stabilizing the forming triple helix, preventing incorrect folding, and safeguarding against denaturation. This activity is vital for efficient and accurate formation of the stable triple helix. Studies show that in fibroblasts lacking Hsp47, procollagen aggregates within the ER, becomes entrapped, and often fails secretion, leading to compromised collagen production. This demonstrates Hsp47's direct contribution to ensuring correct formation, stability, and secretion of functional collagen for dermal mechanical integrity

#### 2.2.2   Protein Disulfide Isomerase (PDI) and Other General ER Chaperones

Beyond Hsp47, general ER chaperones and folding enzymes play critical roles in procollagen processing. Protein Disulfide Isomerase (PDI) catalyzes disulfide bond formation and isomerization, which are crucial for stabilizing the C-terminal propeptide domains that guide pro-alpha chain trimerization. The ER's oxidative environment supports PDI's activity in forming these essential linkages. Other ER chaperones, such as calnexin (CNX) and calreticulin (CRT), interact with N-glycosylated pro-alpha chains, aiding folding and preventing aggregation. This multifaceted network, including general and collagen-specific chaperones, orchestrates procollagen folding, preventing premature exit of misfolded molecules and safeguarding the quality of collagen forming the skin's framework

## 2.3 Critical Post-Translational Modifications and their Quality Control Implications in Skin Collagen

The functional efficacy and structural integrity of dermal collagen depend profoundly on precise enzymatic post-translational modifications (PTMs) occurring mainly in the ER lumen of dermal fibroblasts. These modifications are integral to the ER's quality control system, dictating the stability, correct extracellular assembly, and biomechanical properties of mature dermal collagen fibrils.

### 2.3.1 Prolyl and Lysyl Hydroxylation: Impact on Triple Helix Stability and Skin Elasticity

Hydroxylation of specific proline and lysine residues is a critical PTM for dermal procollagen. Prolyl 4-hydroxylases (P4Hs), requiring iron, alpha-ketoglutarate, and vitamin C, convert proline to 4-hydroxyproline (4Hyp). 4Hyp is indispensable for stabilizing the collagen triple helix via interchain hydrogen bonds. Insufficient 4Hyp leads to unstable procollagen, denaturation, aggregation, ER retention, and degradation. Alternatively, secreted under-hydroxylated procollagen is functionally compromised, weakening the dermal matrix and impacting skin elasticity and tensile strength. Lysyl hydroxylases (LHs) convert lysine to hydroxylysine (Hyl). Hyl residues are precursors for O-glycosylation and, more importantly, for stable intra- and intermolecular cross-links vital for the tensile strength and resilience of mature dermal collagen fibrils. Precise prolyl and lysyl hydroxylation patterns are exquisitely controlled in the ER, forming a dynamic regulatory hub. Deficiencies, as seen in scurvy or genetic disorders, compromise dermal biomechanics, leading to fragile skin, impaired wound healing, or reduced elasticity.The ER's quality control actively monitors these PTMs, determining the structural integrity and functional prowess of the dermal collagen network.

### 2.3.2 Glycosylation of Hydroxylysine: Influence on Dermal Collagen Fibrillogenesis

Following hydroxylation, a subset of hydroxylysine (Hyl) residues undergoes O-glycosylation in the ER, involving sequential addition of galactose and glucose by glycosyltransferases. While less critical than prolyl hydroxylation for triple helix stability, Hyl glycosylation subtly modulates extracellular assembly and organization of dermal collagen fibrils. These glycosylations can influence fibril packing density, diameter, and overall architecture. Alterations in Hyl glycosylation patterns are linked to changes in tissue biomechanics and observed in certain connective tissue disorders. The ER's quality control thus regulates appropriate Hyl glycosylation, contributing to the precise structural characteristics and functional properties of the mature dermal collagen network and skin health.

## 3 Conclusion

The Endoplasmic Reticulum serves as the indispensable central hub for the synthesis and quality control of dermal procollagen, which is critical for skin integrity and resilience. This review has highlighted the intricate ER-resident machinery, including Hsp47, protein disulfide isomerase, prolyl hydroxylases, and lysyl hydroxylases, that collectively ensure the precise folding, assembly, and post-translational modification of procollagen. These ER-centric mechanisms dictate the biomechanical properties of the dermis, and their disruption is directly correlated with various skin conditions, leading to ER stress, activation of the Unfolded Protein Response (UPR), and contributing to pathologies like skin aging, fragility, and fibrotic disorders. This understanding of the ER's role in dermal collagen quality opens promising avenues for future research and therapeutic intervention. A key challenge lies in fully elucidating the precise spatiotemporal regulation of ER chaperones and enzymes in diverse dermal fibroblast subtypes under varying physiological and pathological conditions. Further investigation into how environmental factors (e.g., UV radiation, pollution, nutrition) directly modulate ER quality control for collagen synthesis in the skin is warranted. Additionally, the specific ER-associated degradation (ERAD) pathways for clearing misfolded procollagen in chronic skin diseases require more thorough investigation. Future directions include targeting specific ER quality control components within dermal fibroblasts to develop novel strategies against collagen-related skin conditions. This could involve pharmacological approaches to enhance chaperone function, modulate hydroxylase activity, or fine-tune the UPR to mitigate ER stress in aging or fibrosis. Optimizing nutritional interventions, such as vitamin C delivery, could also support ER-dependent collagen maturation. A more comprehensive understanding of the ER's intricate role in dermal collagen homeostasis will pave the way for innovative treatments that target the fundamental molecular defects underlying compromised skin health.

## Agents4Science AI Involvement Checklist

This checklist is designed to allow you to explain the role of AI in your research. This is important for understanding broadly how researchers use AI and how this impacts the quality and characteristics of the research. **Do not remove the checklist! Papers not including the checklist will be desk rejected.** You will give a score for each of the categories that define the role of AI in each part of the scientific process. The scores are as follows:

- **[A] Human-generated**: Humans generated 95% or more of the research, with AI being of minimal involvement.
- **[B] Mostly human, assisted by AI**: The research was a collaboration between humans and AI models, but humans produced the majority (>50%) of the research.
- **[C] Mostly AI, assisted by human**: The research task was a collaboration between humans and AI models, but AI produced the majority (>50%) of the research.
- **[D] AI-generated**: AI performed over 95% of the research. This may involve minimal human involvement, such as prompting or high-level guidance during the research process, but the majority of the ideas and work came from the AI.

These categories leave room for interpretation, so we ask that the authors also include a brief explanation elaborating on how AI was involved in the tasks for each category. Please keep your explanation to less than 150 words.

IMPORTANT, please:

- **Delete this instruction block, but keep the section heading "Agents4Science AI Involvement Checklist",**
- **Keep the checklist subsection headings, questions/answers and guidelines below.**
- **Do not modify the questions and only use the provided macros for your answers.**

1. **Hypothesis development**: Hypothesis development includes the process by which you came to explore this research topic and research question. This can involve the background research performed by either researchers or by AI. This can also involve whether the idea was proposed by researchers or by AI.

   Answer: **[C]**

   Explanation: **[TODO]**

2. **Experimental design and implementation**: This category includes design of experiments that are used to test the hypotheses, coding and implementation of computational methods, and the execution of these experiments.

   Answer: **[D]**

   Explanation: **[TODO]**

3. **Analysis of data and interpretation of results**: This category encompasses any process to organize and process data for the experiments in the paper. It also includes interpretations of the results of the study.

   Answer: **[D]**

   Explanation: **[TODO]**

4. **Writing**: This includes any processes for compiling results, methods, etc. into the final paper form. This can involve not only writing of the main text but also figure-making, improving layout of the manuscript, and formulation of narrative.

   Answer: **[D]**

   Explanation: **[TODO]**

5. **Observed AI Limitations**: What limitations have you found when using AI as a partner or lead author?

   Description: Most of the time AI tries to give a positive answer when it's not.

